# DIFFender: Diffusion-Based Adversarial Defense against Patch Attacks

## Abstract

Adversarial attacks, particularly patch attacks, pose significant threats to the robustness and reliability of deep learning models. Developing reliable defenses against patch attacks is crucial for real-world applications, yet current research in this area is not satisfactory. In this paper, we propose DIFFender, a novel defense method that leverages a text-guided diffusion model to defend against adversarial patches. DIFFender includes two main stages: patch localization and patch restoration. In the localization stage, we find and exploit an intriguing property of the diffusion model to effectively identify the locations of adversarial patches. In the restoration stage, we employ the diffusion model to reconstruct the adversarial regions in the images while preserving the integrity of the visual content. Importantly, these two stages are carefully guided by a unified diffusion model, thus we can utilize the close interaction between them to improve the whole defense performance. Moreover, we propose a few-shot prompt-tuning algorithm to fine-tune the diffusion model, enabling the pre-trained diffusion model to easily adapt to the defense task. We conduct extensive experiments on the image classification and face recognition tasks, demonstrating that our proposed method exhibits superior robustness under strong adaptive attacks and generalizes well across various scenarios, diverse classifiers, and multiple patch attack methods.

## 1 Introduction

Deep neural networks are vulnerable to adversarial examples (Szegedy et al., 2013; Goodfellow et al., 2014), in which imperceptible perturbations are intentionally added to natural examples, leading to incorrect predictions with high confidence of the model. Most adversarial attacks and defenses are devoted to studying the $\ell_p$-norm threat models (Goodfellow et al., 2014; Carlini & Wagner, 2017; Dong et al., 2018; Madry et al., 2017), which assume that the adversarial perturbations are restricted by the $\ell_p$ norm to be imperceptible. However, the classic $\ell_p$ perturbations require modification of every pixel of the images, which is typically not practical in the physical world. On the other hand, adversarial patch attacks (Brown et al., 2017; Karmon et al., 2018; Li & Ji, 2021; Wei et al., 2022a), which usually apply perturbations to a localized region of the objects, are more physically realizable. Adversarial patch attacks pose significant threats to real-world applications, such as face recognition (Sharif et al., 2016; Xiao et al., 2021), autonomous driving (Jing et al., 2021; Zhu et al., 2023).

Although many adversarial defenses against patch attacks have been proposed in the past years, the defense performance is not satisfactory, which cannot meet the demands of the safety and reliability of real-world applications. Some methods employ adversarial training (Wu et al., 2019; Rao et al., 2020) and certified defenses (Gowal et al., 2019; Chiang et al., 2020), which are only effective against specific attacks but generalize poorly to other forms of patch attacks in the real world (Nie et al., 2022). Another category of patch defense is based on pre-processing techniques (Hayes, 2018; Naseer et al., 2019; Yu et al., 2021; Liu et al., 2022a). They usually destroy the patterns of adversarial patches by image completion or smoothing, but they can hardly restore the images with high fidelity, leading to visual artifacts of the reconstructed images that impact recognition. They can also be evaded by stronger adaptive attacks due to gradient obfuscation (Athalye et al., 2018).

Recently, diffusion models (Sohl-Dickstein et al., 2015; Ho et al., 2020) have emerged as a powerful family of generative models, and have been successfully applied to improving adversarial robustness by purifying the input data (Nie et al., 2022). By diffusing the adversarial examples with Gaussian noises and recovering the original inputs through the reverse denoising process of diffusion models,

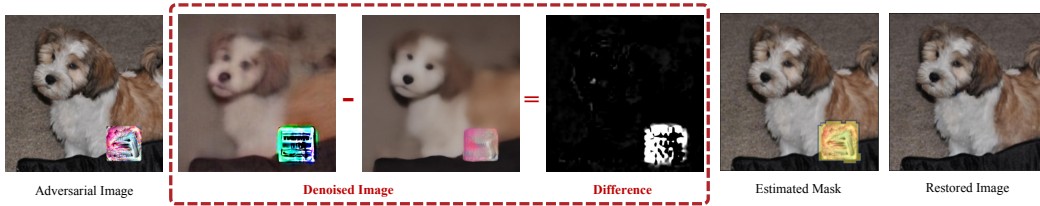

Figure 1: The intriguing property of the diffusion model. We perform a diffusion model multiple times on the given adversarial image, and find the differences between any two various denoised images are pronounced within the adversarial patch regions, which can be used to estimate the mask for this adversarial patch.

the downstream classifiers can correctly recognize the denoised images with high robustness. Our initial intuition is to explore whether diffusion purification can defend against patch attacks. However, we observe that the patch region in the adversarial image can hardly be denoised towards the clean image while the output also differs from the raw patch. But other regions outside the adversarial patch can be denoised more accurately. This indicates that diffusion purification (Nie et al., 2022) is not suitable for purifying adversarial patches since it cannot remove the adversarial patches completely, but also suggests the potential to adopt diffusion models to detect the patch region by comparing the differences between various denoised images, as shown in Fig. 1.

Based on this observation, we propose **DIFFender**, a novel defense method against adversarial patch attacks with pre-trained diffusion models. DIFFender localizes the region of the adversarial patch by comparing the differences between various denoised images and then recovers the identified patch region in the image while preserving the integrity of the underlying content. Importantly, these two stages are carefully guided by a unified diffusion model, thus we can utilize the close interaction between them to improve the whole defense performance. Specifically, we incorporate a text-guided diffusion model such that DIFFender can localize and recover the adversarial patches more accurately with textual prompts. Moreover, we design a few-shot prompt-tuning algorithm to facilitate simple and efficient tuning, enabling the pre-trained diffusion model to easily adapt to the adversarial defense task for improved robustness. The pipeline of DIFFender is illustrated in Fig. 2.

In summary, our contributions are as follows:

- We discover the intriguing property of the diffusion model, that is, disparities exist in the alteration magnitude between the adversarial patch and background regions during the diffusion denoising process, where the difference between any two various denoised images can be further used to identify the adversarial patch.

- Based on the observation, we utilize a unified diffusion model throughout the entire process to localize and restore adversarial patches, and design an effective prompt-tuning algorithm, enabling the diffusion model to easily adapt to the defense task. To the best of our knowledge, this is the first work to defend against patch attacks based on diffusion model.

- We conduct extensive experiments on image classification, face recognition, and further in the physical world, demonstrating that DIFFender exhibits superior robustness under adversarial patch attacks. The results indicate that DIFFender can also generalize well to various scenarios, diverse classifiers, and multiple attack methods.

## 2 RELATED WORK

**Adversarial attacks.** Deep neural networks (DNNs) can be misled to produce erroneous outputs by introducing small perturbations to input examples. Most adversarial attacks (Goodfellow et al., 2014; Moosavi-Dezfooli et al., 2016; Madry et al., 2017; Dong et al., 2018) typically induce misclassification or detection errors by adding small perturbations to the pixels of input examples. However, while these methods can effectively generate adversarial examples in the digital world, they lack practicality in the real world. On the other hand, adversarial patch attacks aim to deceive models by applying a pattern or a sticker to a localized region of the object, which are more realizable in the physical world (Brown et al., 2017; Karmon et al., 2018; Li & Ji, 2021; Wei et al., 2022a; Zhong et al., 2022).

**Adversarial defenses.** With the development of attacks, various defense methods have been proposed. However, most existing defenses primarily focus on global perturbations with $\ell_p$ norm constraints, and defenses against patch attacks have not been extensively studied. Despite the effectiveness of adversarial training (Wu et al., 2019; Rao et al., 2020) and certified defenses (Gowal et al., 2019; Chiang et al., 2020) against specific attacks, they have limited generalization to other patch attacks.

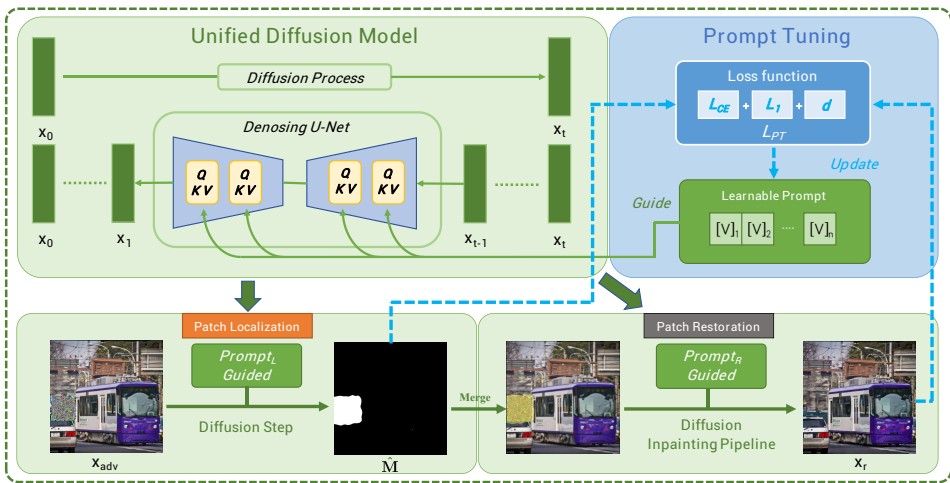

Figure 2: Pipeline of DIFFender. DIFFender leverages a unified diffusion model to jointly guide the localization and restoration of adversarial attacks, and combines a prompt-tuning module to facilitate efficient tuning.

Therefore, most studies focus on pre-processing defenses. Digital Watermarking (Hayes, 2018) utilizes saliency maps to detect adversarial regions and employs erosion operations to remove small holes. Local Gradient Smoothing (Naseer et al., 2019) performs gradient smoothing on regions with high gradient amplitudes, taking into account the high-frequency noise introduced by patch attacks. Feature Normalization and Clipping (Yu et al., 2021) involves gradient clipping operations on images to reduce informative class evidence based on knowledge of the network structure. Jedi (Tarchoun et al., 2023) utilizes entropy to obtain masks. However, these methods can hardly reconstruct the original image and can be evaded be adaptive attacks (Athalye et al., 2018). In contrast, we propose to leverage pre-trained diffusion models to better localize and restore the adversarial patches.

## 3 METHODOLOGY

In this section, we firstly give the intriguing property of diffusion models to localizing patch regions, and then introduce the whole framework of our DIFFender.

### 3.1 OBSERVATION

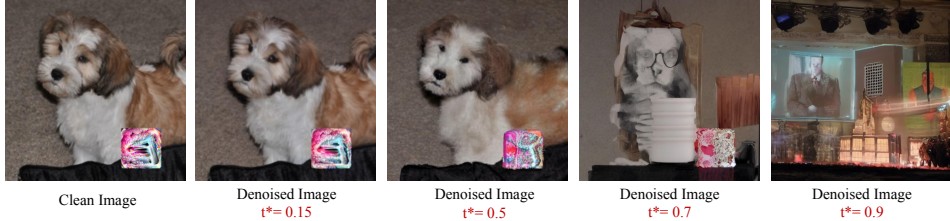

Figure 3: Restoration results denoised by diffusion model at noise ratios $t^* = 0.15/0.5/0.7/0.9$. The patch cannot be removed with small ratios, but the global structure gets lost with large ratios.

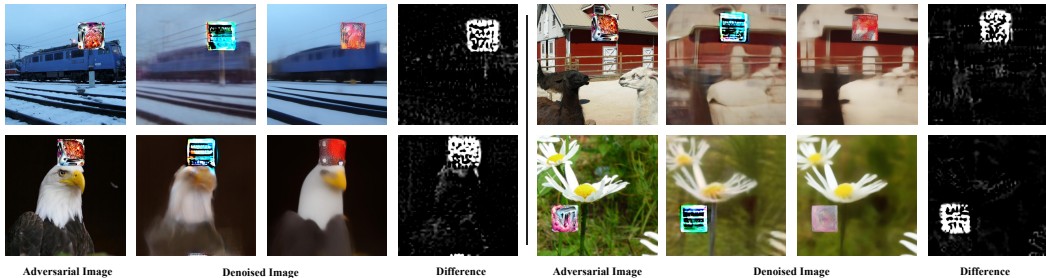

Figure 4: We test the difference results on 100 images from ImageNet and find that differences between any two various denoised images become more pronounced within the adversarial patch regions.

DiffPure (Nie et al., 2022) is a recent method that utilizes diffusion models to remove the imperceptible perturbations on the images by input purification. It first adds Gaussian noise with a noise ratio $t^*$ to the adversarial image and then denoises the image with the backward process of diffusion models, showing good performance. We first examine whether DiffPure can be applied to defending against patch attacks. As illustrated in Fig. 3, using a small noise ratio (e.g., $t^* = 0.15$ as used in DiffPure) is insufficient to remove the patch pattern. It is reasonable since adversarial patches are usually unbounded, such that applying a small Gaussian noise and performing denoising with diffusion models cannot completely remove the patch. Nonetheless, Nie et al. (2022) prove that the clean data distribution and adversarial data distribution get closer over the forward diffusion process, as

$$\frac{\partial D_{KL}\left(p_t \| q_t\right)}{\partial t} \leq 0, \tag{1}$$

where the $KL$ divergence of clean data distribution $p_t$ and adversarial data distribution $q_t$ with noise ratio $t$ monotonically decreases through the forward process. Therefore, using a larger noise ratio (e.g., $t^* = 0.7$ or $t^* = 0.9$), can remove the adversarial patch. However, we find that it changes the image semantics significantly, as show in Fig. 3. This is due to the trade-off between purifying the adversarial perturbations (with a larger $t^*$) and preserving the image semantics (with a smaller $t^*$), making it impossible to find an appropriate noise ratio that can defend against adversarial patches.

Although the results indicate that DiffPure is not directly applicable to adversarial patches, we also find that when the noise ratio is about $0.5$, the patch region in the adversarial image can hardly be denoised but other regions outside the adversarial patch can be well-preserved. It indicates that we can calculate the difference between various denoised image to identify the region of adversarial patch. Four examples about this phenomenon are give in Fig. 4. The reason behind this phenomenon may be that adversarial patches are often meticulously crafted perturbations with a complexity far exceeding the noise present in real image datasets. Diffusion models are trained to learn the probability distribution of real images, thus they struggle to fully adapt to the distribution of adversarial patches, leading to inaccurate estimations.

## 3.2 DIFFENDER

**Patch localization.** DIFFender first performs accurate patch localization based on the above observation of the diffusion model. Given the adversarial image $\mathbf{x}_{adv}$, we first add Gaussian noise to create a noisy image $\mathbf{x}_t$ with noise ratio $0.5$. Next, we apply a text-guided diffusion model to obtain a denoised image $\mathbf{x}_p$ from $\mathbf{x}_t$ with a textual prompt $prompt_L$, and $\mathbf{x}_e$ with empty text. We can estimate the mask region $\hat{\mathbf{M}}$ by taking the difference between the denoised images $\mathbf{x}_p$ and $\mathbf{x}_e$. However, the diffusion model incurs a significant time cost due to the time steps $T$ required. To address this issue, we directly predict the image $\mathbf{x}_0$ from the noisy image $\mathbf{x}_t$ with only one step.

Although the one-step predicted results often exhibit discrepancies and increased blurriness compared to the original image, the differences between one-step predictions still align with our observations. In practice, we perform one-step denoising twice, obtaining two results: $\mathbf{x}_a$, the one guided by $prompt_L$, and $\mathbf{x}_b$, the one guided by empty text to calculate the difference, as:

$$\hat{\mathbf{M}} = \text{Binarize}\left(\frac{1}{m}\sum_{i=0}^{m}(\mathbf{x}_a^i - \mathbf{x}_b^i)\right), \tag{2}$$

where we calculate the difference for $m$ times to enhance the stability and effectiveness, and reduce the time complexity. $prompt_L$ can be hand-designed (e.g., "adversarial") or automatically tuned as shown in Sec. 3.3.

**Patch restoration.** After locating the adversarial patch region, DIFFender then restores the region to eliminate the adversarial effects, while also considering preserving the overall coherence and quality of the image. In particular, we combine the estimated mask $\hat{\mathbf{M}}$ and $\mathbf{x}_{adv}$ as inputs to the text-guided diffusion model with prompt $prompt_R$ to obtain a restored image $\mathbf{x}_r$. We follow the inpainting pipeline in Stable Diffusion (Rombach et al., 2022) to process the mask, where a UNet is used with an additional five input channels to incorporate the estimated mask $\hat{\mathbf{M}}$. Similarly, $prompt_R$ can be manually set (e.g., "clean") or automatically tuned.

**Unified defense model.** It needs to be pointed out that, the above-mentioned two stages are meticulously formulated into one unified diffusion model(e.g., stable diffusion). Therefore, we can take

advantage of the close interaction between these two stages to improve the defense. This also introduces the prompt-tuning module, which involves the joint optimization of the entire pipeline.

## 3.3 PROMPT TUNING

Following the aforementioned pipeline, leveraging visual-language pre-training, DIFFender is capable of efficiently performing zero-shot patch localization. While it is accurate in segmenting adversarial regions in most cases, subtle discrepancies in the segmented masks may occur in certain situations. Given that visual-language pretraining takes advantage of large-capacity text encoders to explore a vast semantic space, to facilitate the effective adaptation of learned representations into our adversarial defense task, we introduce the algorithm of prompt tuning (Zhou et al., 2022).

**Learnable prompts.** First, we replace the textual vocabulary with the learnable continuous vectors. Thus, $prompt_L$ and $prompt_R$ can be transformed into vectors in the following format:

$$prompt_L = [V_L]_1[V_L]_2 \ldots [V_L]_n; \quad prompt_R = [V_R]_1[V_R]_2 \ldots [V_R]_n, \tag{3}$$

where each $[V_S]_i$ or $[V_R]_i$ ($i \in \{1, \ldots, n\}$) is a vector of the same dimensionality as word embeddings. $n$ is a hyperparameter that specifies the number of context tokens, we set $n = 16$ by default. The text content used to initialize $prompt_L$ and $prompt_R$ can be manually provided or randomly initialized.

**Tuning process.** After obtaining the learnable vectors, we design three loss functions for prompt tuning, which jointly optimize the prompt vectors to capture the characteristics of the adversarial regions and improve the defense performance.

First, to accurately identify the adversarial regions for the patch localization module, we employ the cross-entropy loss by comparing the estimated mask $\hat{\mathbf{M}}$ with the ground-truth mask $\mathbf{M}$.

$$L_{CE}(\mathbf{M}, \hat{\mathbf{M}}) = -\sum_{i=1}^{d} \mathbf{M}_i \log(\hat{\mathbf{M}}_i), \tag{4}$$

where $i$ indicates the $i$-th element of the mask. Next, in the patch restoration module, our objective is to restore the mask region while eliminating the adversarial effect of the image. To ensure effective defense, we calculate the $\ell_1$ distance between the restored image $\mathbf{x}_r$ and the clean image $\mathbf{x}$ as

$$L_1(\mathbf{x}_r, \mathbf{x}) = |\mathbf{x}_r - \mathbf{x}|. \tag{5}$$

Lastly, to verify that the adversarial effects have been eliminated, we draw inspiration from Liao et al. (2018) and (Zhang et al., 2018) to make the high-level feature representations of the downstream classifiers between the restored image $\mathbf{x}_r$ and the clean image $\mathbf{x}$ close to each other. Specifically, we compute the $\ell_2$ distance between their feature representations at each layer weighted by a layer-wise hyperparameter as

$$d(\mathbf{x}_r, \mathbf{x}) = \sum_l \frac{1}{H_l W_l} \sum_{h,w} \|w_l \odot (\hat{y}_{rhw}^l - \hat{y}_{chw}^l)\|_2^2, \tag{6}$$

where $l$ denotes a certain layer in the network, $\hat{y}_r^l, \hat{y}_c^l \in \mathcal{R}^{H_l W_l C_l}$ are the unit-normalize results in the channel dimension, and vector $w^l \in \mathcal{R}^{C_l}$ is used for scaling activation channels.

By summing up the aforementioned three losses, we have the overall loss $L_{PT}$ for prompt tuning as

$$L_{PT} = L_{CE}(\mathbf{M}, \hat{\mathbf{M}}) + L_1(\mathbf{x}_r, \mathbf{x}) + d(\mathbf{x}_r, \mathbf{x}). \tag{7}$$

We perform gradient descent to minimize $L_{CE}$ w.r.t. $prompt_L$ and $prompt_R$ for prompt tuning. The design of continuous representations also enables thorough exploration in the embedding space.

**Few-shot learning.** During prompt-tuning, we utilize a minimal number of images for few-shot tuning. Specifically, DIFFender is trained on a limited set of attacked images from a single scenario and a specific attack method, but can still learn optimal prompts that generalize well to other scenarios and attacks, which makes the tuning straightforward and results in a short time requirement.

## 4 EXPERIMENTS

### 4.1 EXPERIMENTAL SETTINGS.

**Datasets and network architectures.** We consider ImageNet(Deng et al., 2009) dataset for evaluation and compare with seven state-of-the-art defense methods: Image smoothing-based defenses,

Table 1: Clean and robust accuracy against patch attacks on ImageNet by Inception-V3 and Swin-S.

| Defense \ Attack | Inception-V3 | | | | | Swin-S | | | | |
|---|---|---|---|---|---|---|---|---|---|---|
| | Clean | AdvP | LaVAN | GDPA | RHDE | Clean | AdvP | LaVAN | GDPA | RHDE |
| Undefended | 100.0 | 0.0 | 8.2 | 64.8 | 39.8 | 100.0 | 1.6 | 3.5 | 78.1 | 51.6 |
| (Dziugaite et al., 2016) | 48.8 | 0.4 | 15.2 | 64.8 | 13.3 | 85.2 | 0.8 | 5.9 | 77.0 | 38.7 |
| (Xu et al., 2017) | 72.7 | 1.2 | 14.8 | 57.8 | 16.4 | 86.3 | 2.3 | 5.5 | 68.8 | 34.8 |
| (Hayes, 2018) | 87.1 | 1.2 | 9.4 | 62.5 | 28.5 | 88.3 | 0.0 | 5.1 | 77.3 | 66.0 |
| (Naseer et al., 2019) | 87.9 | 55.5 | 50.4 | 67.2 | 49.6 | 89.8 | 65.6 | 59.8 | 82.0 | 69.1 |
| (Yu et al., 2021) | 91.0 | 61.3 | 64.8 | 66.4 | 46.5 | 91.8 | 6.3 | 7.4 | 77.0 | 63.7 |
| (Nie et al., 2022) | 65.2 | 10.5 | 15.2 | 67.6 | 44.9 | 74.6 | 18.4 | 26.2 | 77.7 | 62.3 |
| (Tarchoun et al., 2023) | **92.2** | 67.6 | 20.3 | 74.6 | 47.7 | 93.4 | 89.1 | 12.1 | 78.1 | 67.6 |
| DIFFender | 91.4 | **88.3** | **71.9** | **75.0** | **53.5** | 93.8 | **94.5** | **85.9** | **82.4** | **70.3** |

including JPEG (Dziugaite et al., 2016)and Spatial Smoothing (Xu et al., 2017); Image completion-based defenses, such as DW (Hayes, 2018), LGS (Naseer et al., 2019); Feature-level suppression defense FNC (Yu et al., 2021) and Jedi(Tarchoun et al., 2023), a defense based on entropy and diffusion-based defense diffpure (Nie et al., 2022). For classifiers, we consider two advanced classifiers trained on ImageNet: CNN-based Inception-V3(Szegedy et al., 2016) and Transformer-based Swin-S(Szegedy et al., 2016) .

**Adversarial attacks.** We employ AdvP (Brown et al., 2017) and LaVAN (Karmon et al., 2018), which randomly select patch positions and generate perturbations. We also employ GDPA (Li & Ji, 2021) which optimizes the patch's position and content to execute attacks, and RHDE(Wei et al., 2022a), which utilizes realistic stickers and searches for their optimal positions to launch adversarial attacks. To adaptively attack the baseline based on preprocessing, we approximate gradients using the BPDA (Athalye et al., 2018), which implies that the defense methods are white-box against the attack methods and set the number of iterations for the attacks to 100. For adapting the attack on DIFFender, we use an additional Straight-Through Estimator (STE) (Yin et al., 2019) during backpropagation through thresholding operations. Additionally, due to the randomness introduced by the denoising processes, we use the Expectation over Transformation (EOT) + BPDA attack (Athalye et al., 2018).

**Evaluation metrics.** We evaluate the performance of defenses under standard accuracy and robust accuracy. Due to the computational cost of adaptive attacks, unless otherwise specified, we assess the robust accuracy on a fixed subset of 512 sampled images from the test set. To facilitate the observation of changes, we ensure that the selected subset consisted of images correctly classified.

## 4.2 EVALUATION ON IMAGENET

**Experimental results.** Tab. 1 presents the experimental results, where the highest accuracy is highlighted in bold. Based on these results, we draw the following conclusions:

(1)DIFFender outperforms in defense effectiveness. Under strong adaptive attacks utilizing gradients, such as the BPDA+AdvP and BPDA+LaVAN, DIFFender exhibits superior defense performance, even only involving an 8-shot process. This can be attributed to that DIFFender is built upon the unified diffusion framework. Intuitively, the diffusion model can effectively remove adversarial areas while ensuring a high-quality and diverse generation that closely follows the underlying distribution of clean data. Additionally, the inherent stochasticity in the diffusion model allows for robust stochastic defense mechanisms (He et al., 2019), which make it a well-suited "defender" for adversarial attacks.

(2) Image processing defense methods, such as JPEG, SS, and DW, experience a significant decrease in robust accuracy under adaptive attacks. This can be attributed to the algorithms' gradients can be easily obtained. Other methods, such as LGS, FNC, and Jedi, consider the robustness against adaptive attacks. For instance, FNC achieves respectable robust accuracy on Inception-v3. However, its defense effectiveness diminishes when applied to the Swin-S. This is because the feature norm clipping layer proposed is specifically designed for handling CNN feature maps, while DIFFender exhibits excellent generalization capabilities that can extend to different classifiers.

(3)In the experiments, DIFFender only undergoes 8-shot prompt-tuning specifically for the AdvP method. This demonstrates the generalization capability of DIFFender to handle unseen attack methods. For Jedi, it has strong robustness against several attack methods, such as AdvP, but its

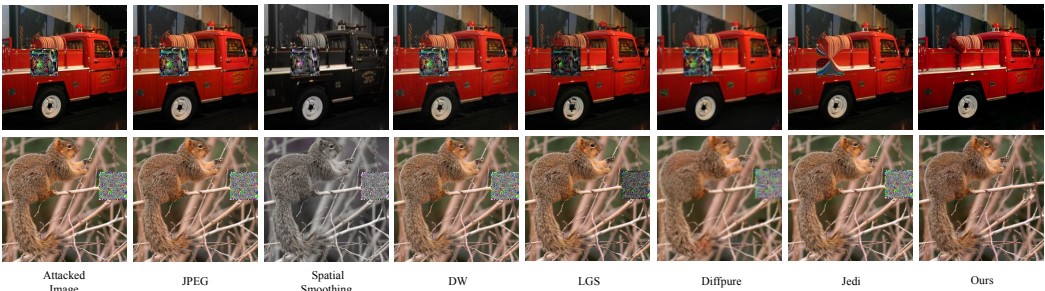

Figure 5: Visualization with examples from ImageNet. The restored images of DIFFender no longer show any traces of the patch, and the restored details are remarkable.

Table 2: Ablation study for different loss functions of DIFFender.

| $L_{CE}$ | $L_1$ | $d$ | Clean | AdvP | LaVAN | GDPA | RHDE |
|---|---|---|---|---|---|---|---|
| | ✓ | ✓ | 91.8 | 76.2 | 66.0 | 72.3 | 49.2 |
| ✓ | ✓ | | 88.3 | 87.1 | 69.5 | 73.8 | 52.7 |
| ✓ | | ✓ | 90.2 | 87.1 | 69.1 | 73.0 | 52.0 |
| ✓ | ✓ | ✓ | **91.4** | **88.3** | **71.9** | **75.0** | **53.5** |

robust accuracy has significantly decreased under other methods, like LaVAN. This might be because the autoencoder used by Jedi is trained under a specific style and cannot generalize well.

(4)When defending against global perturbations with $\ell_p$-norm constraints, diffpure achieves excellent results. However, it performs poorly when facing patch attacks. Specifically, in Tab. 1, when tested against AdvP and LaVAN, the inceptionv3 model purified by diffpure only maintains robust accuracy rates of 10.5% and 15.2%, respectively. This is also consistent with our observations in Section 3.1

**Visualization.** Fig. 5 presents the defense results of the defense methods against patch attacks. Since FNC suppresses the feature maps during the inference stage, it is not shown in Fig. 5. Other methods such as JPEG and DW only exhibit minor changes in the reconstructed images and fail to defend against adaptive patch attacks. After Spatial Smoothing defense, the images show color distortion and are still vulnerable to adaptive attacks. In the case of the LGS method, the patch area is visibly suppressed, which improves the robust accuracy to some extent, but the patch area is not completely eliminated. For Jedi, its patch localization algorithm fails under certain patch attacks, as the second line in Fig. 5 and the restored results cannot achieve complete recovery. On the other hand, the restored images after DIFFender defense no longer show any traces of the patch, and the restored details are remarkable (e.g., the recovery of tree branches in the second column of images).

### 4.3 Ablation studies and additional results

**Impact of loss functions.** To evaluate the effectiveness of different losses, we conduct separate tuning experiments by removing loss functions $L_{CE}$, $L_1$, and $d$ separately, as presented in Tab. 2, where we observe that the robust accuracy significantly decreases when optimizing only the Restoration module without optimizing $L_{CE}$, although it led to an improvement in clean accuracy. On the other hand, removing $L_1$ results in a noticeable decrease in clean accuracy, as images cannot be well restored. Eliminating either the $d$ or $L_1$ loss function causes a slight drop in robust accuracy. Finally, DIFFender which incorporates all three loss functions achieves the highest robust accuracy, which demonstrates the importance of joint optimization for the overall performance of DIFFender.

**Impact of restoration module.** To verify the necessity of restoration, we remove the Patch Restoration step as shown in Tab. 3. Experimental results show that the inclusion of Patch Restoration ensures better DIFFender performance. This is because patches may occasionally obscure crucial areas of an image, resulting in a loss of semantic information. The Restoration step can address this issue by recovering lost semantics, aiding classifiers in overcoming challenging scenarios. Furthermore, longer diffusion steps introduce more randomness, which preserves accuracy against adaptive attacks. Consequently, we conclude that the Patch Restoration step is indeed necessary.

Table 3: Ablation study for different modules in DIFFender. DIFFender(NR) denotes "No Restoration".

| Defense | Inception-V3 | | | | | Swin-S | | | | |
|---|---|---|---|---|---|---|---|---|---|---|
| | Clean | AdvP | LaVAN | GDPA | RHDE | Clean | AdvP | LaVAN | GDPA | RHDE |
| DIFFender (NR) | 86.3 | 57.8 | 44.1 | 69.5 | 48.0 | 88.7 | 68.0 | 35.2 | 78.9 | 69.1 |
| DIFFender | **91.4** | **88.3** | **71.9** | **75.0** | **53.5** | **93.8** | **94.5** | **85.9** | **82.4** | **70.3** |

Table 4: Ablation study for different prompt forms. "EP" and "MP" represent "Empty prompt" and "Manual prompt", two zero-shot versions of DIFFender.

| Defense | Inception-V3 | | | | | Swin-S | | | | |
|---|---|---|---|---|---|---|---|---|---|---|
| | Clean | AdvP | LaVAN | GDPA | RHDE | Clean | AdvP | LaVAN | GDPA | RHDE |
| DIFFender (EP) | 91.3 | 60.7 | 32.8 | 71.1 | 47.0 | 93.2 | 52.2 | 32.0 | 79.3 | 65.7 |
| DIFFender (MP) | 90.8 | 67.1 | 48.7 | 70.3 | 47.8 | 92.7 | 80.0 | 53.4 | 77.0 | 67.6 |
| DIFFender | **91.4** | **88.3** | **71.9** | **75.0** | **53.5** | **93.8** | **94.5** | **85.9** | **82.4** | **70.3** |

**Impact of Prompt Tuning.** In Tab. 4 we compared the complete DIFFender with the "Empty prompt" and "Manual prompt" versions of DIFFender. For DIFFender with manual prompts, we set $prompt_L$="adversarial" and $prompt_R$="clean". The prompt-tuned DIFFender shows a significant improvement in robust accuracy compared to the other two zero-shot DIFFenders. This improvement, despite exposure to only a few attacked images, underscores the effectiveness of prompt tuning.

**Cross-model transferability.** Specifically, we conduct 8-shot prompt-tuning using the AdvP attack on both Inception-V3 and Swin-S. We then test the transferability of learned prompts on CNN-based ResNet50(He et al., 2016) and transformer-based ViT-B-16(Dosovitskiy et al., 2020). The results are presented in Tab. 5. DIFFender keeps high robust accuracy when applied to new classifiers. This demonstrates good generalization capabilities of DIFFender.

## 4.4 EXTENSION IN FACE RECOGNITION.

**Experimental settings.** Facial expressions in human faces introduce a rich diversity, together with external factors such as lighting conditions and viewing angles, making face recognition a challenging task. We conducted experiments on the LFW dataset(Huang et al., 2008), and employed two adversarial patch attacks: RHDE(Wei et al., 2022a) and GDPA (Li & Ji, 2021) mentioned above.

**Experimental results.** The results on the LFW dataset are presented in Tab. 6. DIFFender achieves the highest robust accuracy under both the GDPA and RHDE attacks while ensuring the clean accuracy. It is worth noting that DIFFender is not re-tuned specifically for facial recognition. This further demonstrates the generalizability of DIFFender across different scenarios and attack methods. In contrast, JPEG, SS, and the FNC method obtained low robust accuracies, even below the clean accuracy. This is because in the specific context of facial recognition, the classifier focuses more on crucial local features, and preprocessing the entire image can disrupt these important features. Fig. 6 illustrates the results of DIFFender against face attacks. It can be observed that DIFFender accurately identifies the location of the patch and achieves excellent restoration.

## 4.5 EXTENSION IN PHYSICAL WORLD.

We additionally conduct further experiments in the physical world, where select 10 common object categories from ImageNet and performed two types of patch attacks (meaningful and meaningless)(Wei et al., 2022b). Our approach involves generating digital-world attack results first, then placing stickers on real-world objects in the same positions. We test DIFFender under various conditions, including different angles (rotations) and distances. Qualitative results are depicted in

Table 5: DIFFender transferability performance on ResNet50 and ViT-B-16 for ImageNet.

| Defense | ResNet-50 | | | | | ViT-base | | | | |
|---|---|---|---|---|---|---|---|---|---|---|
| | Clean | AdvP | LaVAN | GDPA | RHDE | Clean | AdvP | LaVAN | GDPA | RHDE |
| Undefended | 100.0 | 0.0 | 14.8 | 73.8 | 37.1 | 100.0 | 1.2 | 2.0 | 76.2 | 52.0 |
| DIFFender | 83.6 | 83.2 | 55.9 | 76.2 | 53.5 | 91.0 | 88.3 | 85.2 | 78.9 | 68.0 |

Table 6: Clean and robust accuracy against patch attacks on LFW.

| Defense | Clean | FaceNet | |
|---|---|---|---|
| | | GDPA | RHDE |
| Undefended | 100.0 | 56.3 | 42.8 |
| (Dziugaite et al., 2016) | 44.1 | 16.8 | 17.8 |
| (Xu et al., 2017) | 19.9 | 8.2 | 3.5 |
| (Hayes, 2018) | 37.1 | 15.2 | 7.2 |
| (Naseer et al., 2019) | 60.9 | 71.9 | 53.5 |
| (Yu et al., 2021) | 100.0 | 39.8 | 39.3 |
| (Tarchoun et al., 2023) | 100.0 | 74.2 | 43.9 |
| DIFFender (EP) | 100.0 | 79.3 | 57.2 |
| DIFFender (MP) | 100.0 | 77.0 | 57.2 |
| DIFFender | **100.0** | **81.1** | **60.7** |

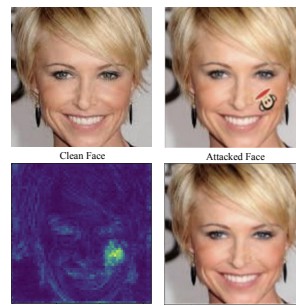

Figure 6: Visualization with examples from LFW attacked by RHDE, localized and restored by DIFFender.

Table 7: Quantitative accuracy of meaningless physical attacks on the Inception-v3 at different angles and distances, along with the quantized results after defense by DIFFender.

| | 0° | yaw ±15° | yaw ±30° | pitch ±15° | distance |
|---|---|---|---|---|---|
| Undefended | 28.9 | 34.8 | 41.8 | 36.7 | 35.9 |
| DIFFender | **80.9** | **76.6** | **77.7** | **75.4** | **73.8** |

Fig. 7 and additional results can be found in Appendix A.3, while quantitative results are presented in Tab. 7, where each configuration is based on 256 frame successfully classified images from the 10 objects selected. Based on the results, we see that DIFFender manifests substantial defensive capabilities across various physical alterations, maintaining its efficacy in real-world scenarios.

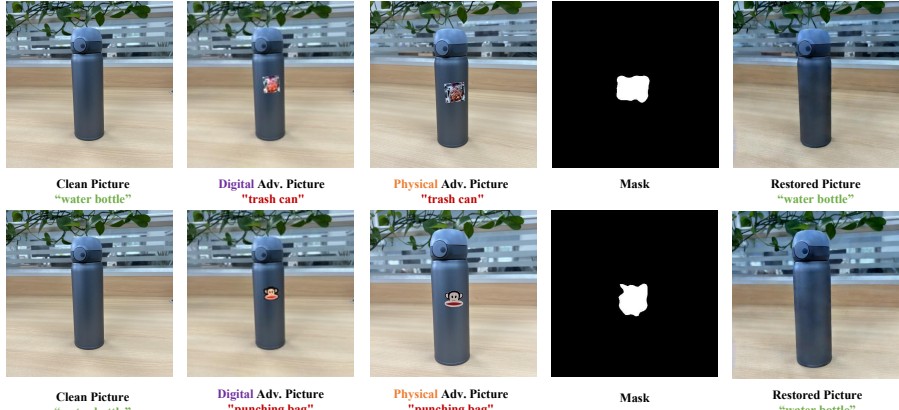

Figure 7: DIFFender defense demonstrations against meaningless patch attacks and meaningful patch attacks in the physical world. Noticing the mask edges may extend slightly beyond the patch region, aids in restoring the patch and helps maintain consistency in the restored image.

## 5 DISCUSSION AND CONCLUSION

We propose **DIFFender**, a novel defense method that leverages a pre-trained unified diffusion model to perform both localization and restoration of patch attacks. Additionally, we design a few-shot prompt-tuning algorithm to facilitate simple and efficient tuning. To show the robust performance of our method, we conduct experiments on image classification, further validate our approach on face recognition and finally extend to the physical world. Our research findings demonstrate that DIFFender exhibits superior robustness even under adaptive attacks and extends the generalization capability of pre-trained large models to various scenarios, diverse classifiers, and multiple attack methods, requiring only a few-shot prompt-tuning. We prove that DIFFender significantly reduces the success rate of patch attacks while producing realistic restored images.

## 6 REPRODUCIBILITY STATEMENT

The code for the method in this paper will be open-sourced. It is based on the Stable Diffusion (https://github.com/Stability-AI/stablediffusion), as well as CoOp(https://github.com/KaiyangZhou/CoOp), which is also available freely. The code to replicate the quantitative results will be made available.

## 7 ETHICS STATEMENT

The Stable Diffusion Model is trained on extensive data scraped from the web, such as LAION, inheriting the biases therein. Hence, the employment of such models can raise ethical concerns, regardless of whether the textual prompts are deliberately harmful or unintentional. We contend that the open-source algorithms in our research can facilitate a more profound understanding of these issues and assist the community in mitigating such concerns in future endeavors. Our code will be made available under a similar license as Stable Diffusion to ensure responsible and ethical utilization of the developed models and algorithms, addressing both moral and legal considerations.

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

# A   ADDITIONAL EXPERIMENTAL RESULTS

## A.1   THE EFFECTS OF $\theta$.

The process of binarizing the differences to obtain a mask estimation $\hat{\mathbf{M}}$ is a crucial step in DIFFender. As shown in Fig. 8. We compare the cases of $\theta = 1.5$ and $\theta = 2.5$. When $\theta = 1.5$, DIFFender achieves the best robust accuracy, reaching 89.1% against AdvP attacks. However, the clean accuracy decreases slightly to 87.1%. On the other hand, when $\theta = 2.5$, the clean accuracy improves, but the robust accuracy decreases. We can observe that there is a tradeoff between clean and robust accuracy.

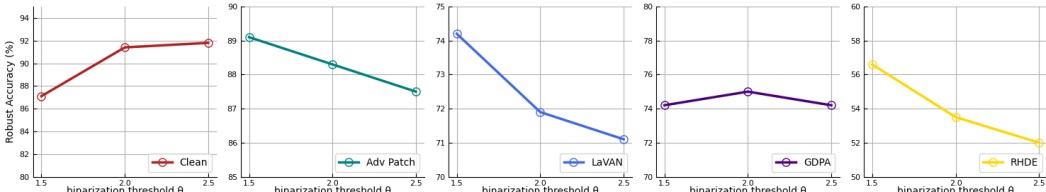

Figure 8: DIFFender performance of different binarization threshold $\theta$ on Inception-v3 for ImageNet. We set $\theta = 2.0$ to showcase the final results.

## A.2   MORE VISUAL RESULTS OF DIFFENDER IN DIGITAL WORLD

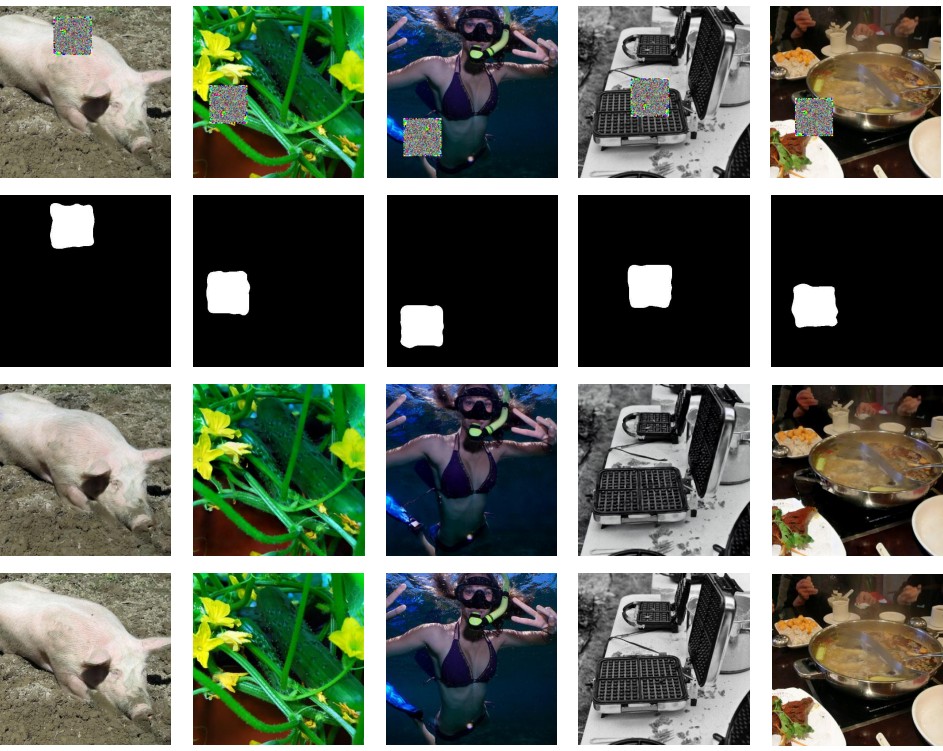

Figure 9: Restoration results of DIFFender. **Row 1:** Images attacked by AdvP. **Row 2:** mask estimation $\hat{\mathbf{M}}$ of the DIFFender Localization module. **Row 3:** Images restored by DIFFender. **Row 4:** The original, non-adversarial images.

Due to space limitations, we only presented a small number of result images in the main text. Here, we showcase additional restoration results from DIFFender, encompassing various scenes, mask prediction outcomes, and comparisons with clean images. The second row in Fig. 9 represents the mask estimation $\hat{\mathbf{M}}$ of the DIFFender Localization module. By prompt tuning, the Localization

module accurately Localizes the patch location and predicts $\hat{M}$ that fully covers the patch region. While the predicted mask edges may occasionally extend slightly beyond the patch region, this aids in restoring the adversarial patch and helps maintain consistency in the restored image.

The third row displays the restored images, while the fourth row shows the original, non-adversarial images. Leveraging the diffusion model, excellent restoration is achieved in various aspects, such as the animal's skin in the first column, plant stems and leaves in the second column, and the structure of tools in the fourth column. This holds true for both grayscale images in the fourth column and color images in the second column. Upon careful observation, one can also notice the accurate rendering of details, such as the subtle shading on human skin in the third column or the metallic reflections and glossiness in the fifth column.

It is evident that the defense results not only eliminate the adversarial patch but also maintain a high level of consistency with the original images. Furthermore, the background remains unaffected, demonstrating the superiority of DIFFender.

### A.3 MORE VISUAL RESULTS OF DIFFENDER IN THE PHYSICAL WORLD.

Fig. 10 and Fig. 11 respectively display more of DIFFender's results in the physical world. To validate the robustness of DIFFender in the physical world, our approach involves initially generating digital-world attack results. Subsequently, stickers are placed on real-world objects in corresponding positions, and images are captured using a camera. These captured images are then inputted to classifiers for analysis.

In Fig. 10, the adversarial patch are created by the AdvP method, while in Fig. 11, the patch attacks are derived from the RHDE method. The third column presents items that were attacked and misclassified in the physical world. The fourth column demonstrates the mask estimation results, and the fifth column displays the images post-defense in each image. It is observable that the adversarial patches are effectively eliminated in the defense results. There are some notable cases; for instance, in the first row of Fig. 10, upon the patch removal, a new pattern appears on the cup in the restored image. However, this conforms to the attributes of a cup and does not interfere with the classifier's categorization. Meanwhile, in the third row of Fig. 11, the method additionally estimates two small regions beyond the adversarial patch. However, these regions are well-restored during the restoration phase, without impacting the final defense or classification results. These results exhibit the practicality of the DIFFender method in the physical world.

## B   MORE DETAILS OF EXPERIMENTAL SETTINGS

### B.1   COMPUTATION COMPLEXITY

All experiments were conducted on an NVIDIA A100 80GB PCIe GPU. During prompt-tuning, we performed 50 rounds of tuning on 8 images attacked with the Adversarial Patch, taking approximately 2 hours for each classifier (inception-v3 and swin-s). For adaptive attacks, we conducted 100 iterations of querying on a fixed subset of 512 sampled images from the ImageNet test set, which took approximately 64 hours using a single GPU. Therefore, running all experiments, including ablation studies and real-world experiments, required approximately 100 GPU days. In Sec.3.2 of the paper, we described an acceleration technique proposed during the Patch Localization stage, which involves directly predicting the result $t_0$, and employed Denoising Diffusion Implicit Model (DDIM) Song et al. (2020) for implementing the diffusion denoising step, utilizing it in both the localization and restoration phases. Those improvement renders our method significantly more efficient, several times over, compared to diffpure, which is also based on diffusion. Further acceleration can be achieved by adopting advanced diffusion variants with accelerated sampling (Liu et al., 2022b; Lu et al., 2022a;b), which we leave for future work.

### B.2   DATASET

In our experiments, we selected a fixed subset of 512 sampled images from the ImageNet test set (also for the LFW dataset) to evaluate our method. We chose the subset based on the following criteria: 1) To facilitate the comparison of adversarial defense methods, we ensured that the selected subset consists of images correctly classified by classifiers, allowing the visual recognition models

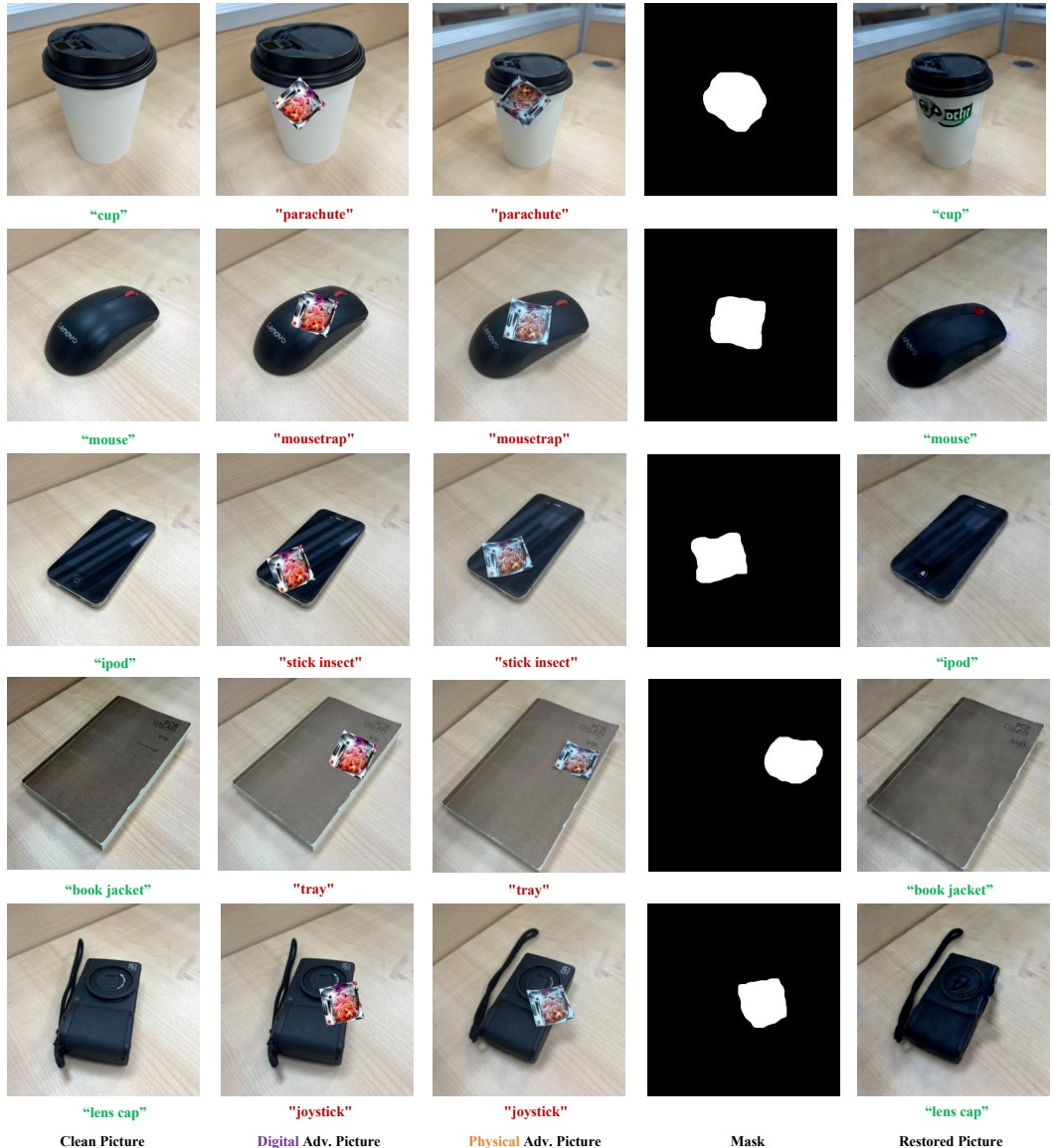

Figure 10: Defense demonstrations against meaningless patch attacks (AdvP) in the physical world. Noticing the mask edges may extend slightly beyond the patch region, aids in restoring the patch and helps maintain consistency in the restored image.

to classify them accurately in their clean state. 2) The subset includes a diverse range of scene categories, including humans, animals, cars, road signs, etc., to validate the method's generalization across different object categories.

Access terms (https://image-net.org/download) state: "Researcher shall use the Database only for non-commercial research and educational purposes. And researcher may provide research associates and colleagues with access to the Database provided that they first agree to be bound by these terms and conditions." Since this work is not used for commercial purposes, we do not violate the license agreement.

Nevertheless, we believe that DIFFender represents the first application of the diffusion pipeline for patch attacks and has been tested against adaptive attacks. This work explores the potential and

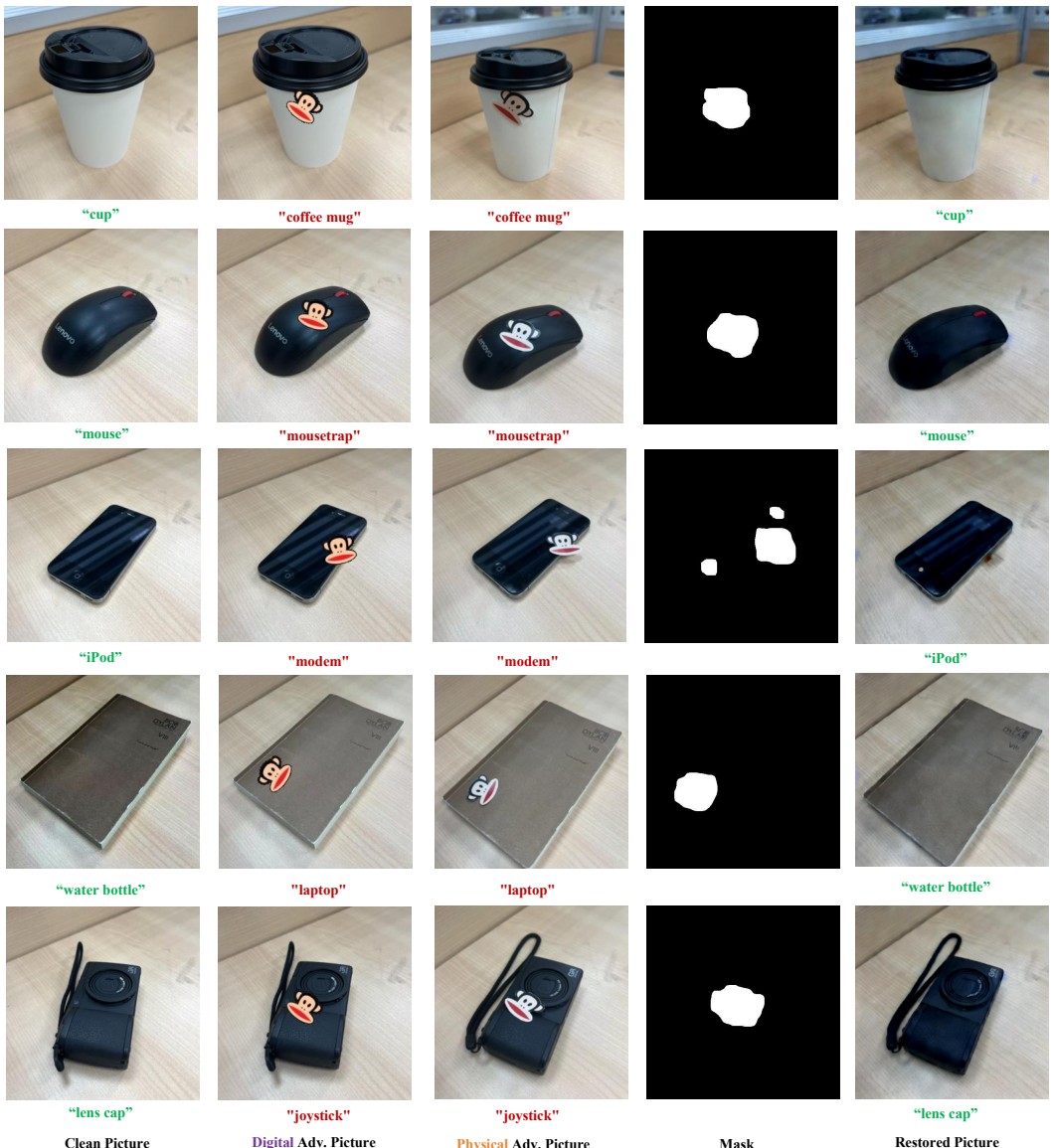

Figure 11: Defense demonstrations against meaningful patch attacks (RHDE) in the physical world.

advantages of using the diffusion model for adversarial defense, an area that has received limited attention. We will continuously enlarge the dataset in the future.

### B.3 IMPLEMENTATION DETAILS OF MASK GENERATION.

In Sec.3.2, we introduced the method by which DIFFender locates patch regions and generates masks. In practical implementation, we obtain the differences from $m$ subtractions and average them. Subsequently, Gaussian smoothing and dilation operations are performed sequentially to optimize the estimated mask results.

Regarding the threshold $\theta$, we calculate the difference between the latent results of denoising for each pair of noisy inputs. The absolute differences of the latent variables are then summed across channels, averaged over multiple channels, and normalized. Afterward, the obtained values are thresholded using the threshold $\theta$, resulting in the final mask estimation $\hat{\mathbf{M}}$.

Since the binarization process is non-differentiable due to the nature of thresholding operations, we utilize the Straight-Through Estimator (STE) during back propagation. This allows us to handle the non-differentiable thresholding and ensure the use of strong adaptive attacks for testing DIFFender.

