# OpenReview forum: "DIFFender: Diffusion-Based Adversarial Defense against Patch Attacks"
_ICLR.cc/2024/Conference — ICLR 2024 Conference Withdrawn Submission_

### Official Review · Reviewer_Xs1L · 2023-10-31

**Soundness:** 3 good
**Presentation:** 3 good
**Contribution:** 2 fair
**Rating:** 5
**Confidence:** 5

**Summary:**

This paper explores empirical defense of adversarial patches. Because the current patch defense is not very satisfactory, the authors proposed DIFFender based on the diffusion model. The authors took advantage of the properties of the diffusion model to enable DIFFender to first locate the location of the patch. Next, DIFFender uses diffusion model image inpainting to eliminate adversarial patches. In order to make the diffusion model adaptively adjust, the authors also introduced a few-shot prompt-tuning algorithm to fine-tune the diffusion model. Extensive experiments illustrate the effectiveness of this approach.

**Strengths:**

1. It is valuable to study the defense methods of adversarial patches.
2. It is interesting to combine the diffusion model with patch defense.

**Weaknesses:**

1. Training of diffusion models requires additional data. This seems to introduce extra data compared to other patch defenses, is this a fair comparison?

2. Patch localization is not very related to adversarial patches. Given any logo posted on it, can it achieve similar positioning? This requires more experimentation.

3. The description of adaptive attacks in Section 4.2 is unclear. We don’t quite understand how adaptive attacks work? In addition, the diffusion model can be backpropagated, which seems to allow white-box attacks, but this is not considered in this paper. This part can refer to the recent DIFF-PGD[1].

4. The motivation for Prompt Tuning is unclear. At present, it is said that the position of the patch can be adaptively positioned, but there is a lack of comparison before and after. There is no relevant experiment in the abaltion study of main body to show that tuning without this prompt is effective. This requires further explanation.

[1] Diffusion-Based Adversarial Sample Generation for Improved Stealthiness and Controllability, NIPS 2023

**Questions:**

See Weaknesses.

---

### Official Review · Reviewer_xWTm · 2023-10-31

**Soundness:** 2 fair
**Presentation:** 2 fair
**Contribution:** 2 fair
**Rating:** 5
**Confidence:** 3

**Summary:**

This paper tackles the problem of adversarial robustness to patch-based attacks. To this end, the authors propose an intricate mechanism to first identify the patch and then remove it. Since the latter part merely refers to inpainting, I believe the main proposal of this paper is to identify an adversarial patch in natural images with the help of diffusion models.

The authors propose to utilize diffusion models in a mechanism consisting of giving a diffusion with a text prompt and again with an empty prompt to compute the difference between the two resulting images to determine the mask. The rest of the methodology seeks to automate this process with prompt tuning.

**Strengths:**

The authors perform several experiments with ablations to help showcase the robustness of their methodology.

**Weaknesses:**

The paper lacks clarity in several aspects. I found it difficult to understand why the method helps in finding the adversarial patch. Although the authors perform elaborate experiments, the method is not well motivated as to its reasoning. I have written more specific concerns in the section on `Questions`.

**Questions:**

Method:
1. Given a denoising step with a text prompt and another step with an empty prompt, why does the difference between the two images have a higher difference at the patch? An intuitive explanation would have been helpful.
2. How is the binarization for the mask (in Eq. 2) performed. I would assume there may be cases where masks are erroneously found at more than one location in the image.
3. In Eq. 2, the difference is computed `m times`, but it is not clear to me why this is performed m times and also what `m` was used in the experiments.
4. The prompt tuning mechanism was very hard for me to understand. Specifically, it would have been great if there was more detail in Subsection 3.3 under `Learnable prompts`. Are prompt_l and prompt_r jointly learned here or in the next step (Tuning process) or both?
5. In Subsection 3.3 under `Tuning process.`, it is not clear why the authors chose l1 distance while for the features, l2 was chosen.
6. In Subsection 3.3 under ` Few-shot learning.` , it is not clear to me how many images were used for this process?

Experiments:
1. For results in Table 1, InceptionV3 and SwinS have been used, which were published in 2016. More recent architectures could have made a stronger case.
2. In Table 4, “Empty prompt" is not explained. And also why is this ablation important?
3. Table 5 and Table 7 do not have any comparison with benchmark methods. These are not specifically ablation studies and hence would have been good to see results from competing methods too.

Question to the authors:
I am trying to disentangle two aspects: prompt tuning and DIFFENDER.
Why cant prompt tuning (for prompt_l) alone help DiffPure recover the adversarial patch? In that case, one does not need to follow Eq. 2 to get the mask?

---

### Official Review · Reviewer_2k2w · 2023-11-01

**Soundness:** 2 fair
**Presentation:** 3 good
**Contribution:** 2 fair
**Rating:** 5
**Confidence:** 4

**Summary:**

This paper proposes a framework for making deep learning models robust against a special type of adversarial example known as a patch attack. It leverages the flexibility of pretrained diffusion models in two steps: Step 1 involves identifying the area of the input under attack (i.e., the patch attack), and in Step 2, restoring (or repainting) this area in such a way that the rest of the input remains unaffected. Firstly, the authors have shown that the original purification methods using diffusion models are unable to recover the area of the patch attack. Subsequently, they argue that they can exploit this fact to localize the affected area using a couple of one-step diffusion processes. Later, they discuss using the same model to fill out the identified attack area. Furthermore, they have demonstrated the effectiveness of adapting prompt-tuning to improve their results in localizing and restoring the attacks.

**Strengths:**

Overall, the proposed methods utilizing pretrained models seem interesting, and they showed intriguing results. They provided different experimental results to demonstrate the usability of their framework, and the presentation of the paper is acceptable, which can be improved with a little bit of polishing.

**Weaknesses:**

However, the proposed method is quite complex and consist of multiple components that require careful calibration. This raises concerns regarding computational cost and gradient obfuscation issues, especially due to the non-differentiability and randomness of components.

At this time, I am inclined to reject the paper, but I remain open to changing my evaluation upon addressing these concerns.

Concerns:
1. As mentioned in the paper, due to the computational cost of performing the attack, you have only tested a subset of 512 images, which is not sufficient to draw a conclusion about the superiority of your method. I would like to understand which part of your method takes more time, whether it is the localization step or the restoration step. I have tested repainting methods, and it takes approximately 7 minutes to repaint a small area in a single photo.
2. I would also like to know how good your approximation of the gradient is and why your method does not suffer from gradient obfuscation.
3. Why did you choose text-guided diffusion models in the first place? What happens if we leverage original diffusion models used in recent works to improve adversarial robustness? (Please check out https://robustbench.github.io/)
4. What does the close interaction between two parts (models) of your framework mean, as you mentioned it a couple of times in the paper? I haven't understood why it would be beneficial. (I see a lack of justification for this design choice.) Please show the opposite result when using different models.
5. Regarding the prompt tuning, I would like to see the effect of different kinds of manual prompts other than what you have mentioned in the paper, "adversarial" and "clean." Please choose related or unrelated prompts. Related (“corrupted” and “uncorrupted”), (“low quality”, “high quality”), unrelated (“image”, “image”), try random words as well.
6. Regarding your results tables, "MEAN+STD" is missing, especially since you have only tested on 512 examples.
7. The use of GDPA is unclear, as it seems weaker than other attacks, which, from your presentation in the paper, conveys that it is a stronger attack that optimizes the location of the patch.
8. Would it be necessary to test your method against the following attacks, as I have seen that previous works have been tested on: Masked PGD (MPGD), Masked AutoPGD (MAPGD), Masked Carlini-Wagner (MCW)?
9. "Cross-model transferability" does not seem surprising, as you have already mentioned that manual prompting works well, so the same prompt settings work for other models.
10. The provided code is not complete, and implementation is missing; also, some parts have syntax errors.

Things that also can be fixed:
1. There are more papers that use diffusion models for purification; please check them out and cite them properly in the paper.
2. The first part of the related work is very similar to the introduction section; please consider rewording it.
3. Figure 2 which shows the structure of your framework has not been referred to in the paper.
4. The format of citations is different; sometimes "(" comes without a space between the sentence and the citation.
5. There is a lack of justification (explanation) for each individual loss in section 4.3.

Very important: Section 3.3 (PROMPT TUNING): The citation of "(Zhou et al., 2022)" in your sentence can lead to the argument that you have proposed the "prompt tuning" method!!

**Questions:**

1. Section 3.2 (DIFFENDER)
    1. What is the noise ratio?
    2. Why should you be able to estimate the mask by taking the difference between x_adv and x_p?
    3. What is the comparison of computational cost between predicting the mask using the mentioned two methods?
    4. Why do you need to have two versions with a prompt and empty text?
2. Section 3.3 (PROMPT TUNING)
    1. Where does the ground-truth mask come from?
3. Section 4.1 (EXPERIMENTAL SETTINGS)
    1. Why did you need to use STE for performing an attack?
    2. Why do you need to use EOT + BPDA attack?
4. Section 4.2 (EVALUATION ON IMAGENET)
    1. What is an 8-shot process?
5. Section 4.3 (ABLATION STUDIES AND ADDITIONAL RESULTS)
    1. How did you train the model without the restoration step, and what did you do with the localized area of the attack (zero it out)? If your localization steps work correctly, you should have achieved better results, as previous work, like PatchZero, showed better performance.

---

### Official Review · Reviewer_jfCV · 2023-11-01

**Soundness:** 3 good
**Presentation:** 3 good
**Contribution:** 2 fair
**Rating:** 3
**Confidence:** 3

**Summary:**

They propose a novel defense method for patch-wised adversarial attack based on the diffusion model -- DIFFender. The proposed method first local the adversarial patch region based on the diffusion based process and then restore the adversarial patch to original patch. The proposed method achieve state-of-the-art defense performance under different type of adversarial attacks.

**Strengths:**

1. The method analysis is clear and visualization is helpful.
2. The experiments are solid and comprehensive.
3. The proposed method can remove the obvious adversarial patch and restore image well which shows good defense performance under different patch-based attacks across different model.

**Weaknesses:**

1. Lack of discussion of the adversarial attack under different attack strength.
2. Lack of discussion of multiple patches situation.
3. It more likes an application of DDIM and stable diffusion.

**Questions:**

1. How does the proposed method perform when the adversarial patch is imperceptible? Can DIFFender still local the patch well?
2. If there are multiple adversarial patches, can DIFFender still local the patch and restore image well?
3. If the important part of the image masked, can DIFFender restore the image well? For example, if a cat face masked, can it restore it well?